# GCL loss in BRAO

**Thomas R. Shearer** [1]*, **Peter N. Steinkamp**[2], **Maria Parker**[2], **Mitsuyoshi Azuma**[1,3]

**1** Department of Integrative Biomedical & Diagnostic Sciences, Oregon Health & Science University, Portland, Oregon, United States of America, **2** Department of Ophthalmology, Oregon Health & Science University, Portland, Oregon, United States of America, **3** Senju Laboratory of Ocular Sciences, Portland, Oregon, United States of America

* Shearert@ohsu.edu

**Data Availability Statement:** All relevant data are within the paper and its Supporting Information files.

**Funding:** TRS, SRA-17-092, Senju Pharmaceutical Co., Ltd, no.

## Abstract

### Purpose

Our recent publication used optical coherence tomography (OCT) to follow thinning of the retinal ganglion cell layer (GCL) in central retinal artery occlusion (CRAO). Thinning of the inner layers also occurs in patients with branch retinal artery occlusion (BRAO). The mechanism for such thinning may be partially due to proteolysis by a calcium-activated protease called calpain. Calpain inhibitor SNJ-1945 ameliorated the proteolysis in a past series of model experiments. The purposes of the present retrospective study were to: 1) use segmentation analysis of OCT images to follow the loss of retinal layers in BRAO compared to CRAO patients, and 2) predict the number of patients and days of observation needed for a clinical trial of a calpain inhibitor against BRAO.

### Methods

A retrospective, case control study was conducted by computer-aided search in a medical records database for BRAO (ICD10 Code H34.239) with at least one OCT procedure (CPT: 92134). Non-proliferative, co-morbid eye diseases were allowed in the patient data base, and manual correction of auto-segmentation errors was performed. GCL thickness changes were followed over time and Cohen-d/sample size statistics were used to predict minimal patients needed for drug trials.

### Results

The thickness of the GCL layer in BRAO decreased rapidly with time as in CRAO, but in more limited quadrants. The data, as fit to a single-phase decay curve, showed that GCL thickness could be used to provide sample size statistics in a clinical trial to test a calpain inhibitor. For example, a 60-day trial with a 60% effective inhibitor would need a minimum of 29 patients.

### Conclusions

Using thickness changes in the GCL layer to monitor the efficacy of potential inhibitors against BRAO and CRAO is practical in human trials requiring a reasonable number of patients and relatively short trial period.

**Competing interests:** I have read the journal's policy and the authors of this manuscript have the following competing interests:TRS, SRA-17-092, Senju Pharmaceutical Co., Ltd, no. MA is an employee of Senju; MP and PS are paid consultants.

## Translational relevance

Measurement of GCL thickness would be a useful indicator of amelioration of BRAO and CRAO progression in a clinical trial of a putative inhibitor.

## Introduction

Branch retinal artery occlusion (BRAO) is characterized by acute painless, monocular vision loss affecting a sector of the visual field (Fig 1A, red arrow). BRAO is caused by obstruction of the arterial branches arising from the central retina artery. Localized regions of superficial retinal whitening are observed along the sector of the retina supplied by the occluded branch [1–3]. The most common causes are carotid or cardiac emboli [1]. The temporal hemisphere is mostly commonly affected and accounts for about over 90% of cases [2]. Even though BRAO is a common ocular vascular occlusive disorder, the incidence rate is less than that of central retinal artery occlusion (CRAO). Among reported cases of acute retinal arterial obstruction,

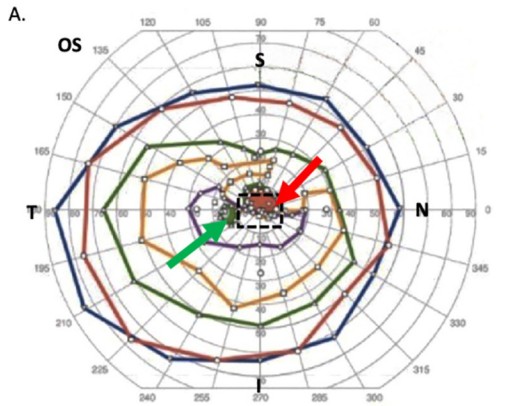
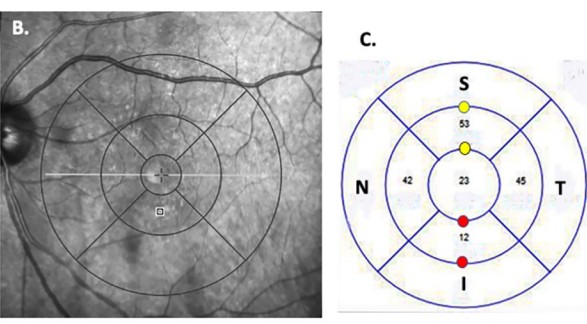
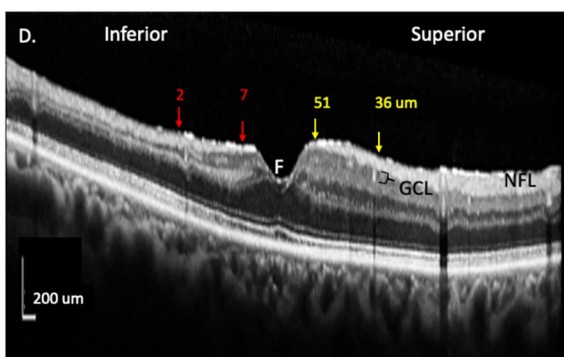

**Fig 1. BRAO in 71-year-old BRAO patient.** A) Visual field exam. Scotoma immediately above the fovea (red, arrow), perceived in the left eye; the green arow indicates the physiological blind spot. The dotted rectangle shows the relative size of a typical optical coherence tomography (OCT) image used in the present experiment, compared to the much larger visual field circumscribed by the blue line. B) SD-OCT (Heidelberg, Germany) with superimposed ETDRS grid from the left eye of the same patient 91 days after BRAO, with sparsely filled arterioles observed in the inferior arcade. (C) After manual segmentation, average ganglion cell thickness values were thinner in the inner, inferior quadrant (12 µm) compared to the temporal, superior, and nasal quadrants. The red and yellow circles indicate measuring points on the edges of the 3 mm inner ring corresponding to the red and yellow arrows in the next image. (D) Measurement of the GCL layer with calipers on a photograph of a vertical b-scan. Image was rotated horizontally, stretched vertically, and integrity of the measurement bars maintained. Bracket indicates the GCL layer below the hyperreflective macular nerve fiber layer (NFL). Numbers above are GCL thicknesses at the arrow heads (red = inferior BRAO region; yellow = normal GCL thickness equidistant from the fovea (F) in the superior quadrant). These point values were roughly similar to the quadrant averages in Fig 1C, confirming the validity of segmentation methodology.

BRAO and cilioretinal artery obstructions account for 38% and 5%, respectively [2]. The visual acuity (VA) in patients with BRAO is much better at the initial and the final visits compared to CRAO [3]. Initial VAs of 20/40 or better occurred in 74% of permanent BRAO cases and 94% of transient BRAO cases, and final visits were VA 20/40 or better in 89% and 100% cases [4–6]. In contrast, patients with poor initial VA of 20/100 or worse do not show significant improvement [7].

Our recent publication showed that OCT could be used to follow thinning of the retinal ganglion cell layer (GCL) in CRAO patients [8]. The data were used to predict the minimum number of patients and observations times needed for clinical trials of a putative drug against CRAO. A consequence of poor arterial flow and hypoxia in retina is accumulation of calcium in RGC cells and activation of a class of intracellular calcium-activated proteases, called calpains [9]. A calpain inhibitor, SNJ-1945, ameliorated changes in the retinal nerve fiber layer (RNFL) and GCL due to hypoxia in monkey and human retinal transplants [10]. This inhibitor can be administered orally and is currently undergoing human clinical trials.

Therefore, the purpose of the studies described below was to use OCT to follow GCL thinning in BRAO and to predict the number of patients and observations times needed for clinical trials of SNJ1945 against BRAO. The importance of proper use of statistical methods was recently noted in a large-scale survey of global visual health [11].

## Materials and methods

The research followed the tenets of the Declaration of Helsinki and OHSU IRB protocol # 15821 (consent waived). A retrospective, case control study was conducted by computer-aided search in a medical records database for BRAO (ICD10 Code H34.239) with at least one OCT procedure (CPT: 92134). Two groups in this study were BRAO eyes and non-BRAO affected eyes. Since this study used readily available BRAO patients, scans were eliminated if they could not be manually corrected, such as in severe proliferative disease.

For each eye with a SD-OCT scan (Heidelberg Engineering GmbH; Heidelberg, Germany) (Fig 1B), mean thicknesses for the eleven Spectralis retinal layers [12] were recorded in the temporal (T), superior (S), nasal (N), inferior (I) quadrants in the superimposed 3 mm ring of the 1, 3, 6 mm Early Treatment Diabetic Retinopathy Study (ETDRS) grid (Fig 1C).

The most affected inner quadrant in the OCT scans of the BRAO eye was followed over time in days. Repeat OCT exams were performed in the follow-up mode. All OCT scans were evaluated for centering of the ETDRS grid on the fovea. When non-follow up data were used, the fovea ETDRS grid was manually centered on the fovea. Auto-segmentation was evaluated and accepted for control non-BRAO eyes, while auto-segmentation was manually corrected (MP, TS) for the BRAO eyes. When a BRAO patient did not have a healthy control eye, the quadrant-specific, mean, layer thickness values determined from a group of 9 non-BRAO eyes were used (Fig 2). These values are similar to previous studies [8].

Results in our study were expressed as % thickness lost in the BRAO eye compared to the non-affected eye, with the "-1 "multiplier added to indicate loss of thickness:

$$\% \textbf{ Thickness change} = (-1) * (1 - \textbf{BRAO}/\textbf{Non}-\textbf{BRAO}) * 100$$

BRAO thickness changes over time were fit by non-linear regression (Prism, GraphPad, ver.8) to a single-phase decay curve:

$$\% \textbf{ Change} = (\textbf{Yo} - \textbf{Plateau}) * e^{(-k*\textbf{DAYS})} + \textbf{Plateau}$$

Yo = % change at time zero, Plateau = final % change, k = decay constant, and e = Euler's constant (2.72).

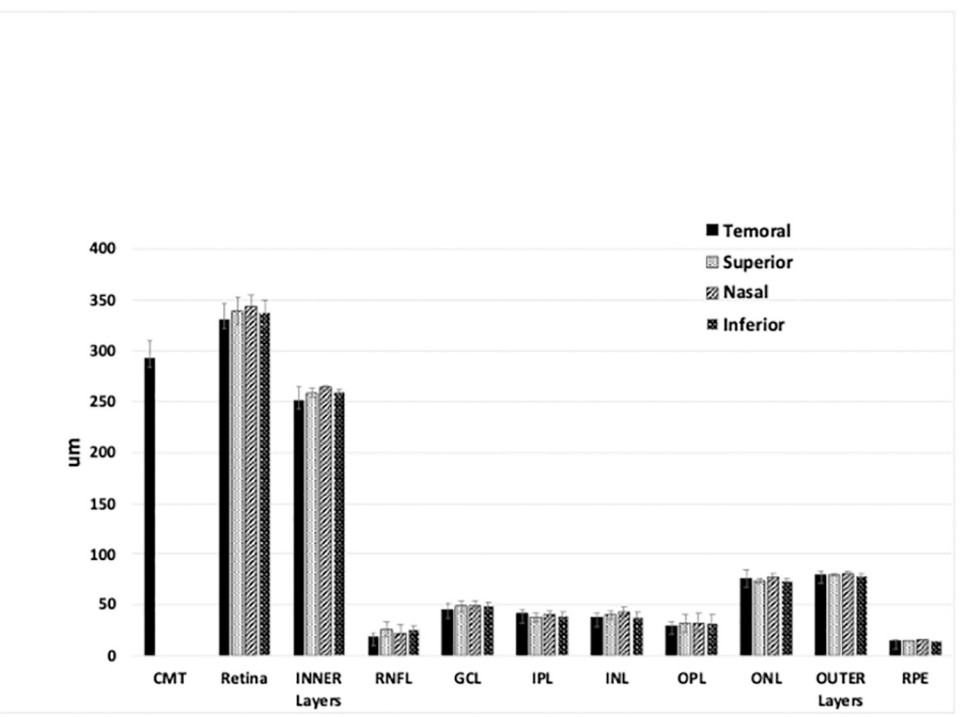

**Fig 2. Normal retinal layer thicknesses in the 3 mm quadrants from non-BRAO eyes (mean ± std. dev. error bars).**
Average age = 64.9 ± 6.3 years (9), all non-Hispanic, ratio of male/female 2:1. (See S1 Table for absolute values).

Effect size (Cohen's-d) was calculated from the standard error of the non-linear regression line and the means from the BRAO and non-affected eyes [13]. The minimum number of patients needed in a clinical trial were determined with a sample size calculator (https://www.ai-therapy.com/psychology-statistics/sample-size-calculator) using a two tailed t-test, 0.05 significance level, 0.8 power, and the Cohen-d effect size.

## Results

### Demographics

BRAO data were obtained from 17 patients with an average age at onset of 61.4 ± 14.6 years (range 27–80); 59% were male, and all were non-Hispanic patients. BRAO SD-OCT scans (n = 38) were evaluated by segmentation software and encompassed a time period from 0 to 1040 days from the initiation of BRAO. The number of scans per patient ranged from 1 to 5 depending on recall visits. Visual acuity ranged from 20/20 to 20/60.

### Retinal layer segmentation in BRAO

BRAO-affected patients in this study showed typical signs of BRAO such as partial loss of vision in one eye (blurriness, sectional blind spots) with mild to moderate loss in overall visual acuity, plaque in arterioles, attenuation of vessels, and thinning of the inner retinal layers. Three individual BRAO patients (Fig 3A–3C) demonstrating thickness changes compared to CRAO (Fig 3D) are shown.

Overall, BRAO showed thinning in the inner layers (RNFL, GCL, IPl and INL) while changes in the outer retinal layers (OPL, ONL, outer layers (photoreceptors), and RPE) were variable and not consistent. The concept of BRAO as a disease of inner rather than outer

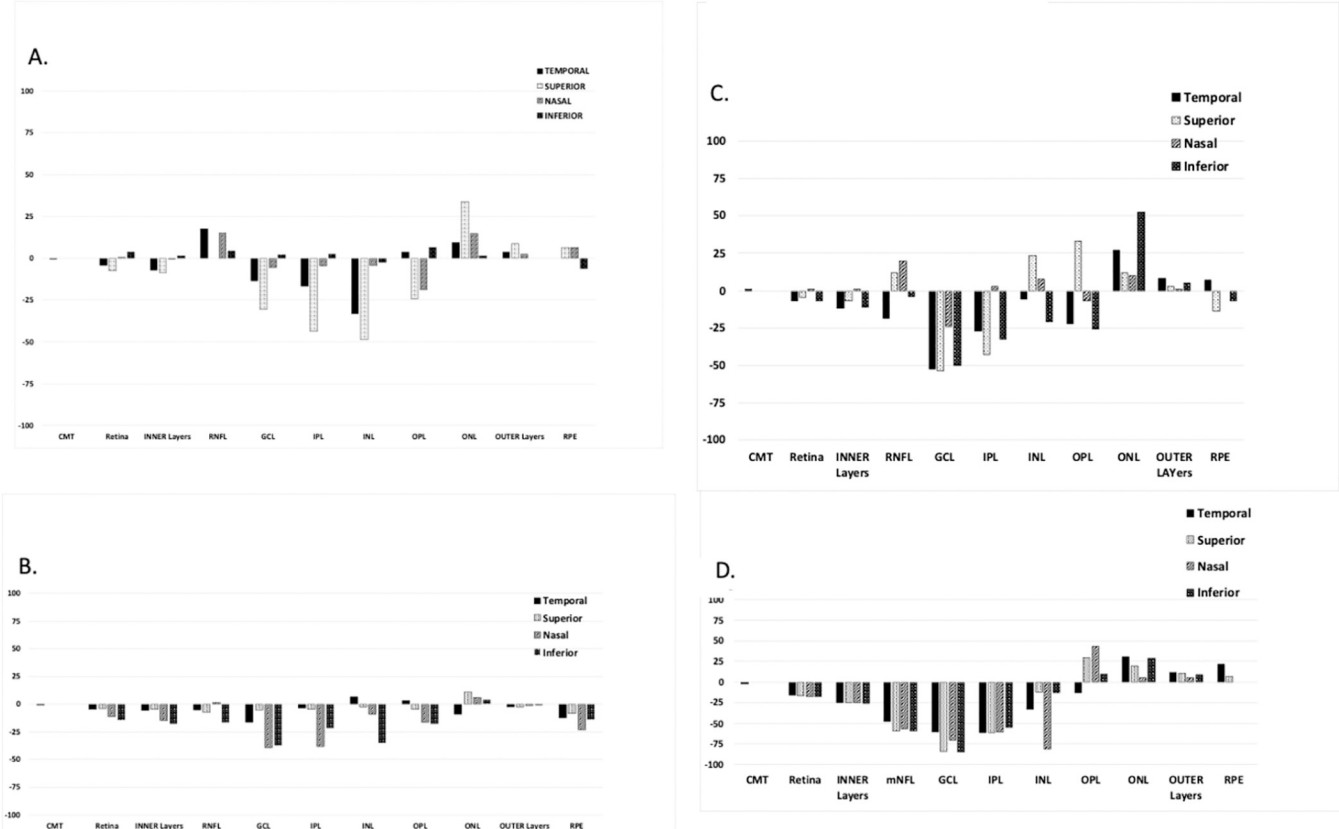

**Fig 3. Retinal layer thickness changes in BRAO and CRAO patients.** (A) Left eye after 28 days from a 69-year-old male patient with BRAO and hypertensive retinopathy OU and VA of 20/30. The most affected GCL thickness decrease was observed in the superior, inner (-31%) and outer (-45%, not shown) quadrants. (B) Right eye from a 68-year-old male patient 36 days after celio-retinal artery infarct BRAO with a VA of 20/25 V. The most affected GCL thicknesses in the inner ring were in the nasal (—39%) and inferior (-37%) quadrants. (C) Right eye at 23 days from a 66-year-old male patient with BRAO and non-proliferative diabetic retinopathy and VA of 20/20. Probable plaque was observed at the first branch point of the superior arterioles. GCL decrease in the superior inner quadrant was (-54%), and the nasal and temporal were similarly affected. (D) Right eye at 47 days from 67-year-old patient with CRAO, retinal artery-vein "nicking," and VA in affected eye of 20/200. The pattern of loss of inner retinal layers was similar to BRAO, but is generally more severe with more quadrants involved.

retinal layers is similar to early studies using OCT to follow BRAO [14]. Since the NFL axons from the GCL bundle to form the optic nerve to the brain, our studies below next focused on the changes over time in the most-affected, inner quadrant of the GCL layer.

## Time course of GCL loss in BRAO

The thickness of GCL layer decreased rapidly with time (Fig 4A). These data significantly fit (P<0.001) to the single-phase decay curve (% Change = $46.2e^{(-0.01*DAYS)}$ - 55.7) with modest goodness of fit at 0.36. The time curves for two individual patients within the data set also fit well to the single-phase decay curve (Fig 4B & 4C). Patient B showed a shallow time course, while patient C showed a steep rapid loss of GCL. These variable time responses contributed to the modest fit of the aggregate data in Fig 4A.

## Sample size estimates

The regression equation above was used to predict GCL changes at specific time points after BRAO (S2 Table) and to calculate effect size (Cohen-D) at different efficacies of a BRAO

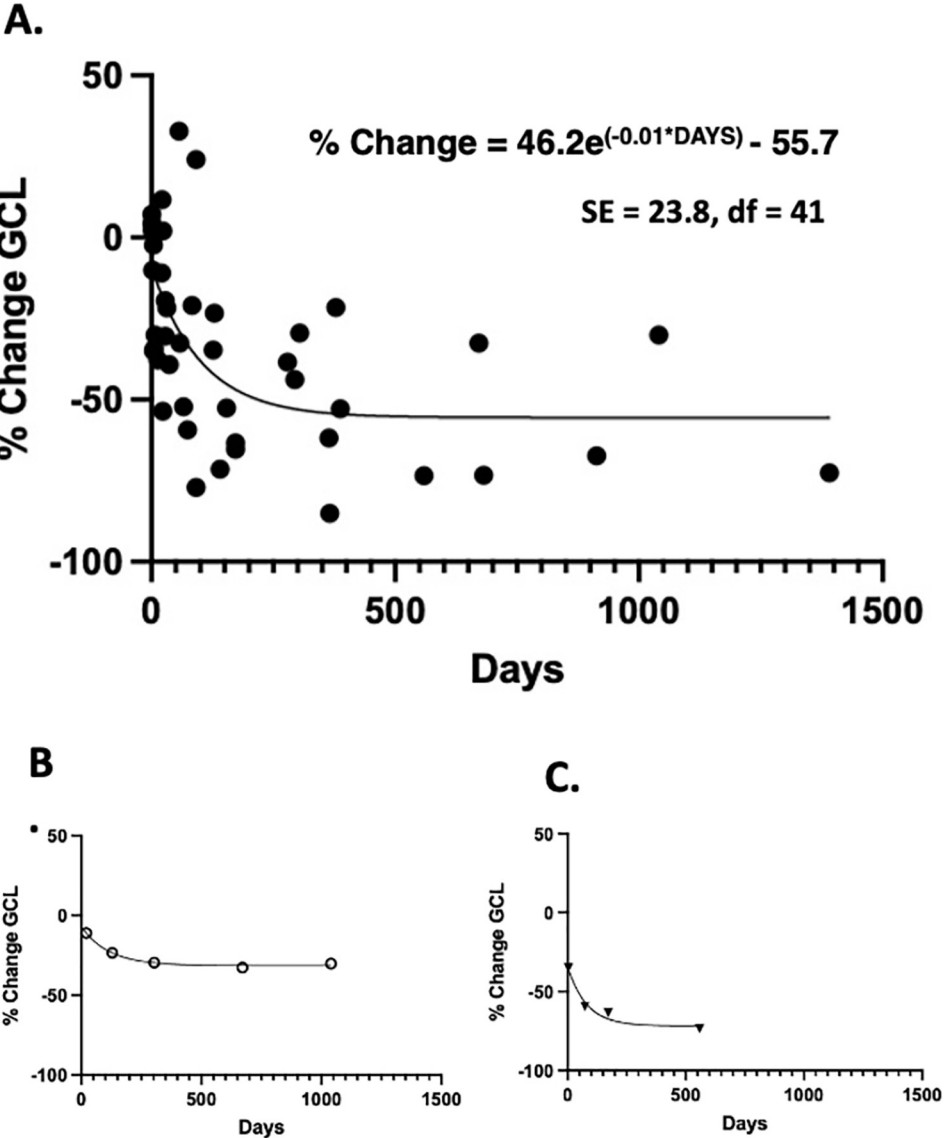

**Fig 4. Time course of GCL thickness loss in BRAO patients.** A). Aggregate data; each filled circle represents a separate OCT workup. B) 73-year-old patient with BRAO in OS. C) 58-year-old patient with BRAO in OD.

inhibitor (S3 Table). From these data, we predicted the number of patients per group needed in a proposed clinical trial of an inhibitor against BRAO (Table 1). For example, a 60-day trial with a 60% effective inhibitor would need a minimum of 29 patients (underlined), assuming one BRAO eye and one unaffected eye per patient. A more effective inhibitor and/or longer trial required fewer patients.

## Discussion

### BRAO clinical trial parameters

This study provided a mathematical description of the time course of GCL loss after BRAO infarct (Fig 4). Using the derived decay formula, subsequent statistical modeling predicted that testing a calpain inhibitor in a clinical trial would be feasible with fewer than 60 patients in a

**Table 1. Minimum number of patients to detect a significant protective effect of a drug against BRAO-induced GCL thickness loss.** Drug efficacies (%, top horizontal row) and trial durations (left vertical column), assuming one BRAO eye and non-affected eye per patient.

| Pt/group | SNJ Efficacy | | | | |
|---|---|---|---|---|---|
| **BRAO Trial** | 20% | 40% | 60% | 80% | 100% |
| **30 Days** | 486 | 130 | 57 | 33 | 21 |
| **60 Days** | 253 | 64 | 29 | 17 | 11 |
| **90 Days** | 165 | 44 | 20 | 12 | 8 |
| **180 Days** | 100 | 26 | 12 | 8 | 6 |

reasonably short time period (Table 1). These practical trial parameters were important to verify because finding adequate number of BRAO patients can be difficult. Such studies are needed since, unfortunately, no consistently effective treatments for BRAO and CRAO currently exist [15]. Both are also considered medical emergencies, where brain magnetic resonance imaging, vascular imaging, and assessment to identify risk for stroke and cardiovascular events are recommended [16].

## OCT as a measure for GCL structural integrity in BRAO

In the current study, auto-segmentation of the retinal layers of non-BRAO eyes by the Heidelberg software worked well. However, future clinical trials with BRAO-affected eyes would be limited by the extensive, time-consuming manual correction [17]. The abrupt changes in reflectivity in the amorphous thinned layers produced aberrant layer thicknesses measurements. In contrast, our CRAO study [8] used a different approach where we first eliminated patients with poorly auto-segmented OCTs, and then used auto-segmentation on the remaining patient eyes. In both cases, segmentation on diseased eyes is difficult. However, clinical trials using OCT to test treatments for BRAO and CRAO are essential since they provide a measure for drug efficacy against loss of structural integrity of the important GCL.

## Complicating factors in BRAO clinical trials

The time course of the GCL decay curve of was variable (Fig 4, $R^2$ = 0.35), potentially complicating interpretation of clinical trials. The reasons for this variability include: acute vs chronic BRAO, the extent to which arterioles were blocked with plaque, the location of blockage along the arteriole arcades, number of quadrants affected, peripheral or macular involvement, reperfusion, perception of scotoma by patients before seeking treatment, and co-morbidity from other eye diseases. Eliminating the eyes with other diseases may not be practical because of the advanced age of many BRAO patients. Indeed, in the present study most patients showed at least one co-morbidity factor such as diabetes, atherosclerosis, hypertension, and/or macular degeneration.

Compared to the major vision loss in CRAO (8), the more sector-specific vision loss in BRAO is another complicating factor. While many BRAO patients have good overall Snellen visual acuity, the smaller scotomas cause loss of quality of life in some patients. The ultimate aim of trials is to verify that a drug preserves visual function. Microperimetry has been used to relate scotoma, capillary drop out, and OCT findings in BRAO [18]. Future clinical trials would benefit from incorporating microperimetry to measure the effect of calpain inhibition on visual acuity. Only one of our patients had both visual field testing and OCTs, which showed a rough association between GCL loss and scotoma (Fig 1A and 1C). Microperimetry could more easily correlate loss of GCL and the smaller scotomas in BRAO over treatment time.

While expected, our study emphasized loss of the GCL in specific quadrants of BRAO eyes compared to the GCL loss in CRAO. This is related to the more limited loss of higher order arterioles in BRAO compared to occlusion of the central retinal artery in CRAO. A calpain inhibitor, such as SNJ-1945, may be effective in both diseases if it were able to penetrate to retinal layers affected by the occlusion and prevent damage by calcium-activated calpain proteolysis. Cytoplasmic calpains have been ubiquitously found in most animal tissues including human retina and GCL [19]. The mechanism for intracellular accumulation of calcium for activation in GCL is unknown. Mechanisms could include generalized permeabilization of the outer cell membrane by initial swelling from the infarct, enhanced calcium ion influx by various ion channels and pores, and efflux of calcium into the cytoplasm from oxygen starved mitochondria. Knowledge of compromised calcium entry points in BRAO could enhance drug therapy by providing specific target sites [18].

## Supporting information

**S1 Table. Retinal thicknesses in the quadrants of the inner (3 mm) and outer (6 mm) rings from non-BRAO eyes (n = 8 or 9).**
(TIF)

**S2 Table. Predicted change in GCL layer during BRAO.**
(TIF)

**S3 Table. Effect size (Cohen-D) used for determining samples sizes.**
(TIF)

## Acknowledgments

The authors sincerely thank Mr. George Petricek generous help with OCT.

## Author Contributions

**Conceptualization:** Thomas R. Shearer, Mitsuyoshi Azuma.

**Data curation:** Thomas R. Shearer.

**Formal analysis:** Thomas R. Shearer, Peter N. Steinkamp, Maria Parker.

**Funding acquisition:** Thomas R. Shearer, Mitsuyoshi Azuma.

**Investigation:** Thomas R. Shearer.

**Methodology:** Peter N. Steinkamp.

**Supervision:** Thomas R. Shearer.

**Validation:** Peter N. Steinkamp.

**Writing – original draft:** Thomas R. Shearer.

**Writing – review & editing:** Thomas R. Shearer, Peter N. Steinkamp, Maria Parker, Mitsuyoshi Azuma.

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
