## [Decision Letter · Decision Letter 0]

1 Dec 2022

PONE-D-22-15899GCL loss in BRAOPLOS ONE

Dear Dr. Shearer,

Thank you for submitting your manuscript to PLOS ONE. After careful consideration, we feel that it has merit but does not fully meet PLOS ONE’s publication criteria as it currently stands. Therefore, we invite you to submit a revised version of the manuscript that addresses the points raised during the review process.

We look forward to receiving your revised manuscript.

Kind regards,

Shiying Li, MBBS

Academic Editor

PLOS ONE

Journal Requirements:

2. Thank you for providing the following Funding Statement: 

“I have read the journal's policy and the authors of this manuscript have the following competing interests:TRS, SRA-17-092, Senju Pharmaceutical Co., Ltd, no. MA is an employee of Senju; MP and PS are paid consultants”

We note that one or more of the authors is affiliated with the funding organization, indicating the funder may have had some role in the design, data collection, analysis or preparation of your manuscript for publication; in other words, the funder played an indirect role through the participation of the co-authors.

If the funding organization did not play a role in the study design, data collection and analysis, decision to publish, or preparation of the manuscript and only provided financial support in the form of authors' salaries and/or research materials, please review your statements relating to the author contributions, and ensure you have specifically and accurately indicated the role(s) that these authors had in your study in the Author Contributions section of the online submission form. Please make any necessary amendments directly within this section of the online submission form.  Please also update your Funding Statement to include the following statement: “The funder provided support in the form of salaries for authors [insert relevant initials], but did not have any additional role in the study design, data collection and analysis, decision to publish, or preparation of the manuscript. The specific roles of these authors are articulated in the ‘author contributions’ section.”

If the funding organization did have an additional role, please state and explain that role within your Funding Statement.

Please also provide an updated Competing Interests Statement declaring this commercial affiliation along with any other relevant declarations relating to employment, consultancy, patents, products in development, or marketed products, etc. 

Additional Editor Comments (if provided):

Based on your previous study which showed OCT could be used to follow thinning of the GCL in CRAO, the authors here found that the thickness of the GCL layer in BRAO decreased rapidly with time as in CRAO, and GCL thickness could be used to provide sample size statistics in a clinical trial to test a calpain inhibitor. But there are still some minor questions needed to address.

1. In ‘Introduction’, since ‘the purpose of the studies described below was to use OCT to follow GCL thinning in BRAO and to predict the number of patients and observations times needed for clinical trials of SNJ1945 against BRAO’, it would be better to add some information about the patients number prediction and observation times for clinical trials from other researches.

2. At the end of the ‘Materials’, last sentence, add ‘.’ and the software you used.

3. In ‘Results-Demographics’, since the ‘co-morbidity factor’ existed, it would be better to be discussed in the discussion.

Reviewers' comments:

Reviewer's Responses to Questions

**Comments to the Author**

1. Is the manuscript technically sound, and do the data support the conclusions?

Reviewer #1: Yes

2. Has the statistical analysis been performed appropriately and rigorously? 

Reviewer #1: Yes

3. Have the authors made all data underlying the findings in their manuscript fully available?

Reviewer #1: Yes

4. Is the manuscript presented in an intelligible fashion and written in standard English?

Reviewer #1: Yes

5. Review Comments to the Author

Reviewer #1: Based on their previous study which showed OCT could be used to follow thinning of the GCL in CRAO, the authors here found that the thickness of the GCL layer in BRAO decreased rapidly with time as in CRAO, and GCL thickness could be used to provide sample size statistics in a clinical trial to test a calpain inhibitor.

1. In ‘Introduction’, since ‘the purpose of the studies described below was to use OCT to follow GCL thinning in BRAO and to predict the number of patients and observations times needed for clinical trials of SNJ1945 against BRAO’, it would be better to add some information about the patients number prediction and observation times for clinical trials from other researches.

2. At the end of the ‘Materials’, last sentence, add ‘.’ and the software you used.

3. In ‘Results-Demographics’, since the ‘co-morbidity factor’ existed, it would be better to be discussed in the discussion.

6. PLOS authors have the option to publish the peer review history of their article (what does this mean?). If published, this will include your full peer review and any attached files.

Reviewer #1: No

---

## [Author Response · Author response to Decision Letter 0]

5 Dec 2022

1. Reviewer 1 comment: …… it would be better to add some information about the patient number prediction and observation times for clinical trials from other researches. Author response and revision: Last sentence presents this and is newly referenced.

2. Reviewer 1 comment: At the end of the ‘Materials’, last sentence, add ‘.’ and the software you used. Author response and revision: The “.” was added at the end of the sentence, and the software was indicated in the middle of the sentence. 

3. Reviewer 1 comment: In ‘Results-Demographics’, since the ‘co-morbidity factor’ existed, it would be better to be discussed in the discussion. Author response and revision: As requested the co-morbidity sentence was removed from the “Demographics” section and moved to the 3rd paragraph of the Discussion.

Additional journal requirements: 

4. Style requirements

The entire revised manuscript was updated to PLOS ONE formatting style requirements

5. Updated funding statement. “I have read the journal's policy and the authors of this manuscript have the following competing interests: TRS, contract SRA-17-092, Senju Pharmaceutical Co. Ltd. MA is an employee of Senju. MP, TS and PS are paid consultants. The funder provided support in the form of salary for author [MA], but did not have any additional role in the study design, data collection and analysis, decision to publish, or preparation of the manuscript. The specific roles of the authors are articulated in the ‘author contributions’ section. This does not alter our adherence to PLOS ONE policies on sharing data and materials.” 

6. Review your reference list to ensure that it is complete and correct, indicating changes. 

References were updated and conform to style requirements. 

Uploaded three files labeled: 

“Response to Reviewers.”

“Revised Manuscript with Track Changes,” The marked-up manuscript highlighting changes made to the original version labeled 

“Manuscript,” the unmarked revised manuscript.

---

## [Editor Report · Decision Letter 1]

19 Dec 2022

GCL loss in BRAO

PONE-D-22-15899R1

Dear Dr. Shearer,

We’re pleased to inform you that your manuscript has been judged scientifically suitable for publication and will be formally accepted for publication once it meets all outstanding technical requirements.

Kind regards,

Shiying Li, MBBS

Academic Editor

PLOS ONE

Additional Editor Comments (optional):

All the comments are responsed.
---

## [Editor Report · Acceptance letter]

22 Dec 2022

PONE-D-22-15899R1 

GCL loss in BRAO 

Dear Dr. Shearer:

I'm pleased to inform you that your manuscript has been deemed suitable for publication in PLOS ONE. Congratulations! Your manuscript is now with our production department. 

Kind regards, 

on behalf of

Dr. Shiying Li 

Academic Editor

PLOS ONE